# Effects of Processing Method and Parameters on the Wall Thickness of Gas-Projectile-Assisted Injection Molding Pipes

**DOI:** 10.3390/polym15091985

**Published:** 2023-04-22

**Authors:** Tangqing Kuang, Jiamin Wang, Hesheng Liu, Zhihuan Yuan

**Affiliations:** School of Mechatronics & Vehicle Engineering, East China Jiaotong University, Nanchang 330013, China

**Keywords:** gas-projectile-assisted injection molding, wall thickness, process method, processing parameter

## Abstract

Gas-Projectile-Assisted Injection Molding (G-PAIM) is a new injection molding process derived from the Gas-Assisted Injection Molding (GAIM) process by introducing a projectile to it. In this study, the short-shot method and the overflow method of both the G-PAIM and GAIM processes were experimentally compared and investigated in terms of the wall thickness of the pipes and its uniformity. The results showed that the wall thickness of the G-PAIM molded pipe was thinner and more uniform than that of the GAIM molded pipe, and the wall thickness of the pipe molded by the Gas-Projectile-Assisted Injection Molding Overflow (G-PAIM-O) process was the most uniform. For the G-PAIM-O process, the influence of processing parameters, including melt temperature, gas injection delay time, gas injection pressure, melt injection pressure and mold temperature, on the wall thickness and uniformity of the G-PAIM-O pipes were studied via the single-factor experimental method. It was found that the effects of gas injection delay time and gas injection pressure on the wall thickness of the G-PAIM-O pipes were relatively significant. The wall thickness of the pipes increased with the increase in gas injection delay time and decreased with the increase in gas injection pressure. The melt temperature, melt injection pressure and mold temperature had little effect on the wall thickness of the G-PAIM-O pipes. In general, the wall thickness uniformity of the G-PAIM-O pipes was slightly affected by these processing parameters.

## 1. Introduction

Fluid-Assisted Injection Molding (FAIM) is a kind of process derived from the conventional injection molding [1,2,3,4]. A pressured fluid medium is introduced to penetrate the polymer melt to assist filling and packing of the melt. The inherent limitations of conventional injection molding are overcome. The products formed by the FAIM process are free from surface shrinkage, warpage and outside bubbles [5,6,7,8,9,10]. According to the fluid medium used, the FAIM process can be divided into Gas-Assisted Injection Molding (GAIM) and Water-Assisted Injection Molding (WAIM) [11,12,13,14]. Compared with conventional injection molding, GAIM has advantages such as material savings, a reduction in residual stresses, more design freedom and the elimination of surface shrinkage of parts [15,16,17,18,19,20]. Compared with WAIM, GAIM has less limitations in its molding materials. However, the hollow section of the FAIM parts is formed by penetration of the fluid medium [21,22], which leads to the product suffering from some defects such as unsmooth inner surfaces and non-uniform wall thickness [13]. 

In 1992, a new injection molding method, named Fluid-Projectile-Assisted Injection Molding (F-PAIM), was proposed in a Japanese patent, which uses water or gas to drive a projectile to penetrate melt and to mold a hollow part [23]. Compared with the FAIM process, more uniform wall thickness and more effective control of wall thickness with less limitation of molding material can be achieved by the F-PAIM process [24,25]. The F-PAIM process can also be categorized into Water-Projectile-Assisted Injection Molding (W-PAIM) and Gas-Projectile-Assisted Injection Molding (G-PAIM) [26]. In view of whether the mold cavity is fully filled with a polymer melt or not before the injection of fluid, the F-PAIM process can be divided into the short-shot method and the overflow method [27]. In the short-shot method, the cavity is first partially filled with a polymer melt and then a pressurized fluid is injected to drive the projectile to penetrate the core of the polymer melt. The shot size of the melt needs to be precisely controlled to obtain the suitable penetration length. This method can be effective in saving material. The overflow method, on the other hand, completely fills the cavity with the polymer. Figure 1 shows a schematic diagram of the overflow method of the F-PAIM process. (1) Firstly, the projectile is placed on the fluid nozzle. (2) Secondly, after the mold is closed, the melt is injected into the mold cavity and over the projectile, and the mold cavity is full of melt. (3) Thirdly, the pressurized fluid is injected from the fluid nozzle after a gas injection delay time, and the projectile is driven through the melt core into an overflow cavity. (4) Finally, the part cools down while the fluid holding pressure compensates for any shrinkage. Then, the fluid is drained before molding opening and the part can be ejected.

Up until now, few research groups besides the Institute of Plastic Processing at RWTH Aachen University (IKV) [26,28,29,30] and East China Jiaotong University [27,31,32,33,34] have been engaged in research on and the promotion of applications of the F-PAIM process. Research on F-PAIM technology has focused on the effect of the processing parameters, the process method and the projectile characteristics of the W-PAIM process on molding quality. Our group conducted numerical simulations of the projectile penetration behavior during the water injection stage of the W-PAIM process for a pipe with an outer diameter of 16 mm. It was found that the wall thickness of a W-PAIM pipe molded with a projectile 12 mm in diameter was much thinner than that of a WAIM pipe. Additionally, when the projectile with a diameter less than 10 mm was used, there was a certain deflection phenomenon during the penetration of the projectile in the melt, and the wall thickness fluctuated greatly [32]. In addition, we also experimentally investigated the effects of the processing method and processing parameters on the wall thickness of the W-PAIM pipes and found that the overflow method could produce a pipe with more uniform wall thickness, and the water injection delay time and melt temperature had significant effects on the wall thickness of the overflow W-PAIM pipes [31]. Compared with W-PAIM, G-PAIM has a wider range for material selection, but there is little relevant research on G-PAIM. This paper aims to investigate the effects of process methods, including Short-shot Gas-Projectile Assisted Injection Molding (G-PAIM-S), Overflow Gas-Projectile Assisted Injection Molding (G-PAIM-O), Short-shot Gas-Assisted Injection Molding (GAIM-S) and Overflow Gas-Assisted Injection Molding (GAIM-O), on the wall thickness and wall thickness uniformity of pipes. Furthermore, the single-factor method was used to investigate the influence of processing parameters on the wall thickness and wall thickness uniformity of the G-PAIM-O pipes. The research results are expected to provide some theoretical guidance for practical production to improve product quality.

## 2. Experimentation

### 2.1. Materials 

The molding material used in the experiments was polypropylene (PP, Grade PPH-T03, China Petroleum & Chemical Corporation, Beijing, China.). It had a melt flow index of 3 g/10 min, a molding shrinkage of 1.65%, a heat deformation temperature of 91 °C and a tensile strength of 32 MPa. The projectile used in the experiments was polyamide 6 (PA6). It has a melting point of 220 °C, a heat deformation temperature of 160 °C and a density of 1.15 g/cm^3^.

### 2.2. Experimental Platform

The experiments were carried out on a lab-developed G-PAIM experimental platform, which comprised an injection molding machine (TTI-250FT, Donghua Machinery Co., Dongguan, China), projectiles, a mold with changeable inserts, an air compressor (V0.6/8, Zhejiang Senlong Electromechanical Co., Taizhou, China), a gas injection unit (GPC-FX-1, Shenzhen Hengrong Gas Assist Equipment Co., Shenzhen, China), a mold temperature controller (BTM-09W, Shenzhen Brake Machinery Co., Shenzhen, China) and gas injection components. The injection molding machine has a maximum injection pressure of 1700 kg/cm^2^. The projectile used in the experiment is shown in Figure 2. The molds are shown in Figure 3. The outer diameters of the straight pipe and the bending pipe molded in the experiments were 16 mm. There was an overflow cavity at the end of the cavity. The switch between the short-shot method and the overflow method could be realized by controlling the sealing pin at the overflow channel connected to the mold cavity with the overflow cavity. The air compressor was a belt-type piston air compressor. The gas injection component consisted of an air injection module and a nozzle.

### 2.3. Experimental Scheme

Firstly, the wall thickness and wall thickness uniformity of a bending pipe molded by the four processes, namely GAIM-O, GAIM-S, G-PAIM-O and G-PAIM-S, were compared. The shot size of the short-shot method was 45% of the cavity volume. The experimental processing parameters were a melt temperature of 230 °C, a gas injection delay time of 5 s, a gas injection pressure of 4 MPa, an injection pressure of 6 MPa and a mold temperature of 25 °C.

Secondly, the single-factor experimental method was used to explore the effects of the processing parameter on the wall thickness and wall thickness uniformity of G-PAIM-O straight pipes. The available processing window was determined by combining the recommendation of material manufacturer and experiments. Then, the processing parameters were set and shown in Table 1 with basic values in brackets. The experiments were carried out with one processing parameter changed and the other processing parameters remaining as basic values.

### 2.4. Characterization of Wall Thickness and Its Uniformity

#### 2.4.1. Measurement of the Wall Thickness of Pipes

Three pipes were taken as measurement specimens for each group of experiments, and the wall thicknesses were measured using digital vernier calipers. The wall thickness measurement scheme is similar for straight and bending pipes. The difference is that only four measuring positions are required for straight pipes, as shown in Figure 4a, whereas five measuring positions are required for bending pipes, as shown in Figure 4b. Take the wall thickness measurement solution for a bending pipe as an example. Five locations, namely P1, P2, P3, P4 and P5, were taken at the straight segments of the pipe and cut transversely, and then, the wall thicknesses at four equal points of each location were measured, as in Figure 4c. The average wall thicknesses of the four equal points at one location were taken as the thickness at that position. The average of the wall thicknesses of the five picked locations on each pipe was the wall thickness of that pipe. Therefore, the average wall thickness of the three pipes of the same group of experiments was taken as the general wall thickness of the group of experimental pipes.

#### 2.4.2. Characterization of Wall Thickness Uniformity

The wall thickness uniformity was also an important indicator of the quality of the pipes. It is characterized by a standard deviation of the calculated wall thickness: the smaller the standard deviation, the more uniform the wall thickness of the pipes. The standard deviation is calculated as follows:(1)$$\sigma_{j}=\sqrt{\frac{\sum_{i=1}^{12n}{\left(t_{i}-t_{j}\right)}^{2}}{12n}}$$
where $t_{i}$ is the wall thickness of the same pipe at position i in the cross-section, $t_{j}$ is the total wall thickness of the j group of pipes, n is the number of measurement positions of the same pipe and $\sigma_{j}$ is the standard deviation of the wall thickness of the j group of pipes.

## 3. Results and Discussion

### 3.1. Effect of Processing Method on the Wall Thickness of Pipes

The average wall thicknesses along the flow direction of the bending pipes molded via the four processes, namely GAIM-O, GAIM-S, G-PAIM-O and G-PAIM-S, are shown in Figure 5.

As can be seen from Figure 5, the wall thickness of the GAIM-O pipes along the melt flow direction decreased from 2.73 mm at P1 to 2.19 mm at P4 and then increased to 2.74 mm at P5. The wall thickness of the GAIM-S pipes along the melt flow direction was reduced from 2.78 mm at P1 to 1.22 mm at P5, with a significant decrease from upstream to downstream. The wall thickness of the G-PAIM-O pipes was more uniform and reduced slightly from 1.46 mm to 1.32 mm along the flow direction. The wall thickness of the G-PAIM-S pipe was reduced from 1.49 mm at P1 to 1.03 mm at P4 along the flow direction, and the projectile did not penetrate to P5.

It can be observed from the results that the wall thicknesses of the G-PAIM pipes were much thinner and more uniform than those of the GAIM pipes for both the overflow method and the short-shot method. This was due to the fact that, in the GAIM process, the high-pressure gas is compressible and more likely to penetrate in the melt with lower viscosity and less resistance at the center of the cavity cross-section, resulting in a smaller gas penetration cross-section and larger wall thickness. Moreover, the size of the gas penetration cross-section is also susceptible to changes in penetration resistance. In contrast, in the G-PAIM process, a solid projectile was driven by the high-pressure gas to penetrate the melt, so the outside diameter of the projectile directly affected the size of the penetration cross-section and the outside diameter of the projectile had little change during the short penetration process.

It was found that the wall thickness of the GAIM-S pipe varied widely along the melt flow direction. This was due to the fact that, in the short-shot process, the melt length at the front end of the gas penetration was shortened as the penetration progressed and the penetration resistance decreased correspondingly, and more melt can be pushed by the gas with a constant pressure, resulting in a larger penetration cross-section. In the GAIM-O process, the diameter of the overflow channel was smaller than that of the cavity, which caused the melt to accumulate near the end of the cavity. Therefore, the wall thickness at the location of P5 was thicker.

The penetration cross-section of the G-PAIM-S pipe was larger than that of the GAIM-S pipe, which means more melt was pushed to the end of the cavity in the G-PAIM-S pipe than that in the GAIM-S pipe. Therefore, with the same shot size, the G-PAIM-S pipes had a shorter penetration length than that of the GAIM-S pipes and the projectile failed to penetrate to the location of P5.

The G-PAIM-O pipes had a more uniform wall thickness in the flow direction. The G-PAIM-S pipes had a more obvious decrease in the wall thickness along the melt flow direction, indicating that the penetration cross-section gradually became larger along the flow direction in the G-PAIM-S process. This was because, in the short-shot process, the melt length at the front of the projectile gradually shortened as penetration proceeded, resulting in a reduction in penetration resistance and a faster advance rate correspondingly. The projectile can drag more melt flow, resulting in a larger penetration cross-section. In the overflow process, the melt that was propelled and displaced during the projectile penetration entered the overflow channel with a smaller cross-section, but the penetration resistance was not significantly reduced. As a result, the penetration speed was smoother, and the penetration cross-section size was correspondingly more stable. Moreover, due to the pushing effect of the solid projectile, the melt accumulated at the end of the cavity was also pushed into the overflow channel.

In Figure 6, the standard deviations of the wall thicknesses of the bending pipes molded by the four processing methods are shown. It can be seen that the standard deviation of the wall thickness of the G-PAIM molded pipe was smaller than that of the GAIM molded pipe, which indicates that the G-PAIM pipe had a more uniform wall thickness. Among the four methods, G-PAIM-O had the smallest standard deviation of 0.068, which indicates that it had the most uniform wall thickness. The GAIM-S standard deviation was the largest at 0.646, which indicates that it had the worst uniform wall thickness.

It can be concluded that, among the four process methods GAIM-O, GAIM-S, G-PAIM-O and G-PAIM-S, G-PAIM-O has the capability to produce pipes with thinner and more uniform wall thickness. Therefore, the G-PAIM-O process was taken as the object in investigating the effects of processing parameters on the wall thickness of its pipe.

### 3.2. Effect of Processing Parameters on the Wall Thickness of G-PAIM-O Pipes

The melt temperature, gas injection delay time, gas injection pressure, melt injection pressure and mold temperature were chosen to explore those influences on the wall thickness of G-PAIM-O straight pipes.

The effects of the melt temperature on the wall thickness and its uniformity are shown in Figure 7. The wall thicknesses of the pipes were reduced from 1.28 mm to 1.25 mm when the melt temperature was increased from 210 °C to 230 °C. Whereas when the melt temperature continued to rise to 240 °C, the wall thickness of the pipes increased slightly (to 1.26 mm). Generally, the melt temperature had little influence on the wall thickness of the G-PAIM-O pipes. In our research on the W-PAIM process, it was found that the projectile penetrated in the melt with a drag flow at the outside of its sidewall, and the projectile dragged the melt flow in the area adjacent to its sidewall [31]. As the melt temperature rose, the melt viscosity gradually decreased. As a consequence, the projectile dragged more melt flows, which resulted in a thinner wall thickness of the pipes. When the melt temperature rose to 240 °C, a temperature higher than the melting point of 220 °C of the projectile material of PA6, the surface of the projectile was partially melted and the effective outer diameter of the projectile became smaller, resulting in a smaller penetration cross-section and a thicker wall thickness of the pipes. It can also be seen from Figure 7 that the standard deviation of the wall thickness decreased first and then increased with the increase in melt temperature. The reason for this may be that, as the melt temperature increased, the viscosity of the melt decreased, which resulted in better flow of the melt. Therefore, the penetration centrality and stability of the projectile were better, resulting in a smaller standard deviation. However, too high a melt temperature resulted in partial melting of the projectile surface during its penetration, which affected the actual outer diameter of the projectile during penetration. Therefore, the standard deviation of the wall thickness increased.

Figure 8 shows the effect of gas injection delay time on wall thickness and its uniformity. With the extension of the gas injection delay time from 2 s to 8 s, the wall thickness of the G-PAIM-O pipes increased from 1.20 mm to 1.32 mm, which was not a negligible effect. The reason for this may be that the cooling time of the mold wall to the melt increased with the increase in gas injection delay time: consequently, the more the melt temperature decreased, the higher the melt viscosity. Accordingly, the resistance of the projectile to penetration increased and less melt could be dragged by the projectile during its penetration, which led to a thicker wall thickness of the molded pipe. Figure 8 shows that the standard deviation of wall thickness gradually increased with the extension of gas injection delay time, which meant that the wall thickness uniformity became worse. The reason for this may be that the increase in gas injection delay time made the cooling time of the melt increase. Therefore, the non-uniformity of cooling became more obvious, and the asymmetry of melt viscosity in the cavity became more significant. As a result, the centrality of the projectile penetration became worse, leading to an increase in the standard deviation. Moreover, the increase in melt viscosity caused an increase in penetration resistance and less stability of the projectile penetration, which led to an increase in the standard deviation.

The results of the effect of gas injection pressure on the wall thickness and its uniformity are shown in Figure 9. When the gas injection pressure increased from 4 MPa to 6 MPa, the wall thicknesses of the pipes were reduced from 1.31 mm to 1.17 mm, with a more obvious effect. However, when the gas injection pressure continued to rise to 7 MPa, the reduction in the wall thickness of the pipes became slower. The reason for this may be that a higher gas injection pressure caused a faster penetration of the projectile, more shear heat generated around the projectile and the melt viscosity decreased. Consequently, the projectile dragged more melt to flow forward, which resulted in a thinner wall thickness. Nevertheless, as the gas injection pressure continued to increase, its effect on the wall thickness of the pipe weakened for the presence of the solidification layer at the mold wall and the limited influence of the projectile drag flow. It can also be found in Figure 9 that the standard deviation of wall thickness increased gradually with the increase in gas injection pressure, which indicated that the uniformity of wall thickness became worse. This may be because the projectile penetration velocity increased with the increase in gas injection pressure. The shear heat on the side of the projectile caused the surface of the projectile to melt partially, which led to a smaller outside diameter of the projectile during penetration. Therefore, the standard deviation increased.

The effect of melt injection pressure on the wall thickness and its uniformity are shown in Figure 10. As the melt injection pressure increased, the wall thickness of the pipes fluctuated in a small range between 1.31 mm and 1.30 mm, and the standard deviation of the wall thickness also decreased slightly. This indicates that the melt injection pressure had less effect on the wall thickness of the pipes: the higher the injection pressure, the more uniform the wall thickness, albeit slightly. This may be due to the fact that, on the one hand, the increased injection pressure led to a denser melt and an increase in melt viscosity. On the other hand, the higher the melt injection pressure, the faster the melt injection rate, which caused the shear to increase and the melt temperature to increase consequentially, both of which resulted in a decrease in melt viscosity. The two aspects offset each other and led to an insignificant effect of melt injection pressure on the penetration of the projectile.

Figure 11 shows the effect of mold temperature on the wall thickness and its uniformity. It can be seen that the wall thickness of the pipes tended to decrease with the increase in mold temperature, from 1.28 mm to 1.22 mm, which is a small decrease. The standard deviation of the pipe wall thickness also tended to decrease with the increase in mold temperature. This was perhaps due to the fact that the increase in mold temperature resulted in a lower temperature difference between the mold and the melt, which led to a thinner solidification layer and a slower increase in melt viscosity. The projectile can drag more melt flow, with better centrality and more stability of its penetration. As a result, the wall thickness was thinner and more uniform.

In general, under all the processing conditions investigated, the standard deviations of the wall thickness of the G-PAIM-O pipes were below 0.12, which meant a relatively uniform wall thickness. This indicated that the influence of processing parameters on the wall thickness uniformity of the G-PAIM-O pipes was not significant.

## 4. Conclusions

The effects of process methods and processing parameters on the wall thickness of pipes molded via the variants of the Gas-Assisted Injection Molding process were experimentally investigated. The following conclusions were drawn.

The wall thicknesses of the G-PAIM pipes were much thinner and more uniform than those of the GAIM pipes for both the overflow and short-shot methods, in which the G-PAIM-O pipes had the most uniform wall thickness. The G-PAIM-S pipe had a larger penetration cross-section and shorter penetration length compared with the GAIM-S pipe with the same shot size. Among the four processes GAIM-O, GAIM-S, G-AIM-O and G-AIM-S, G-PAIM-O had the capability to mold pipes with thinner and more uniform wall thickness.

The influence of the gas injection pressure and gas injection delay time on the wall thickness of G-PAIM-O pipes was significant: the longer the gas injection delay time and the lower the gas injection pressure, the thicker the wall thickness. In contrast, those of the melt temperature, melt injection pressure and mold temperature were minimal. 

Among the four variants of the Gas-Assisted Injection Molding process, the wall thickness uniformity of G-PAIM-O pipes was good and less affected by the processing parameters.

## Figures and Tables

**Figure 1 polymers-15-01985-f001:**
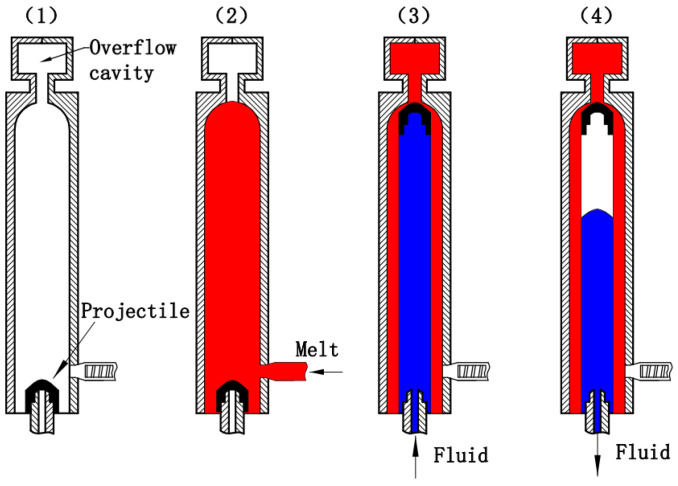
Forming process of the F-PAIM overflow method process.

**Figure 2 polymers-15-01985-f002:**
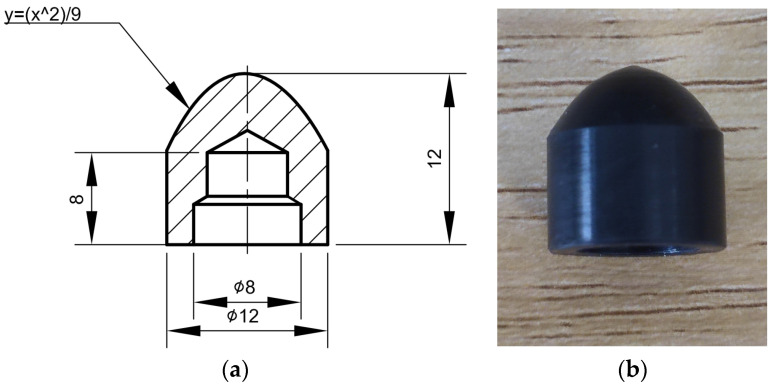
Projectile used in the experiment: (**a**) dimensional drawing of the projectile; (**b**) physical drawing of the projectile (all dimensions are in millimeters).

**Figure 3 polymers-15-01985-f003:**
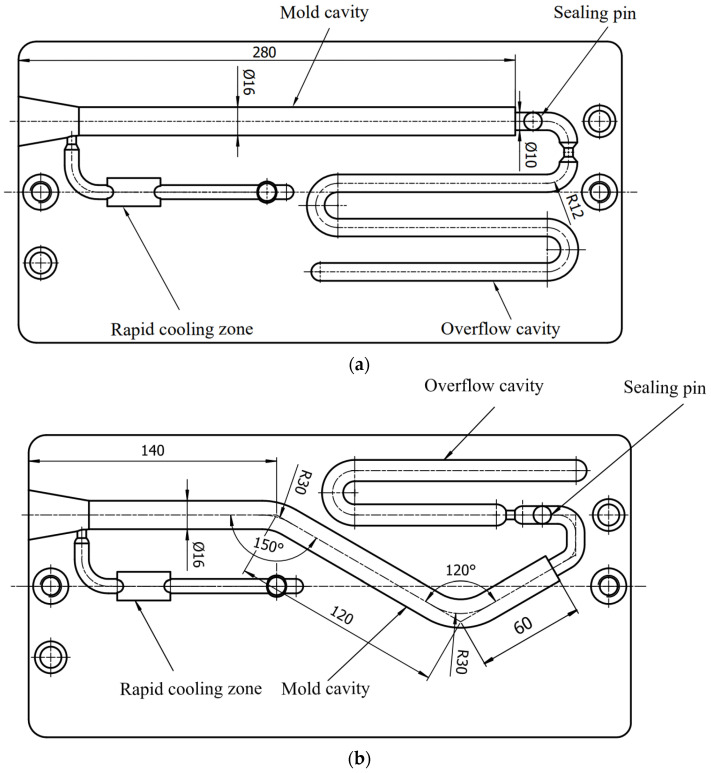
Diagram of mold used in the experiment: (**a**) mold for straight pipe; (**b**) mold for bending pipe (all dimensions are in millimeters).

**Figure 4 polymers-15-01985-f004:**
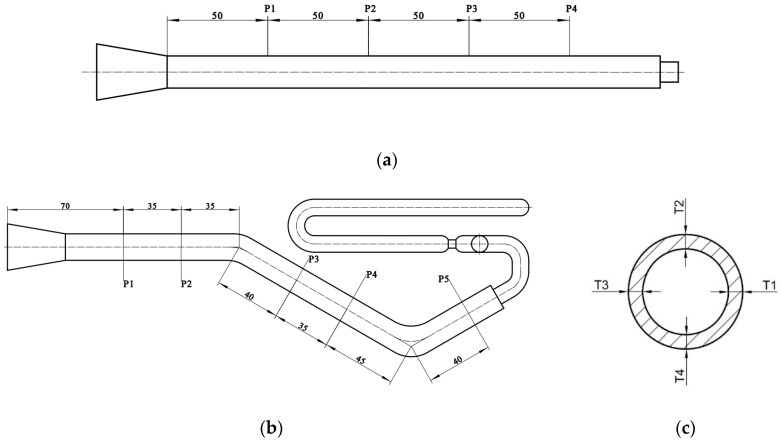
Wall thickness measurement of the pipe: (**a**) measuring positions of the straight pipe; (**b**) measuring positions of the bending pipe; (**c**) measuring points of each section (all dimensions are in millimeters).

**Figure 5 polymers-15-01985-f005:**
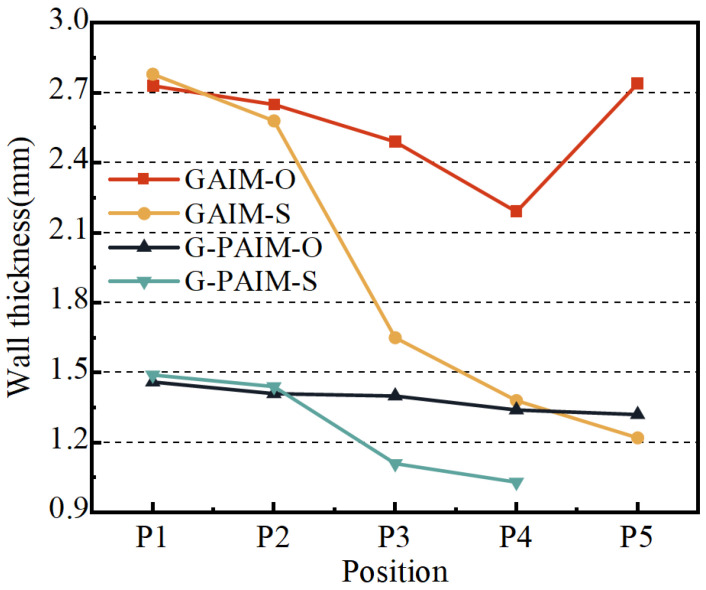
Wall thickness at different positions of pipes molded by four processes.

**Figure 6 polymers-15-01985-f006:**
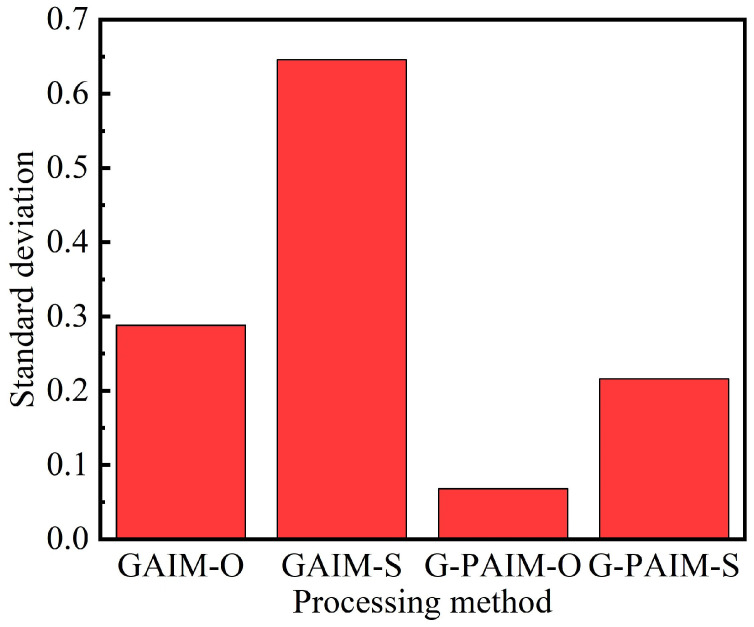
Standard deviation of pipes molded by four processes.

**Figure 7 polymers-15-01985-f007:**
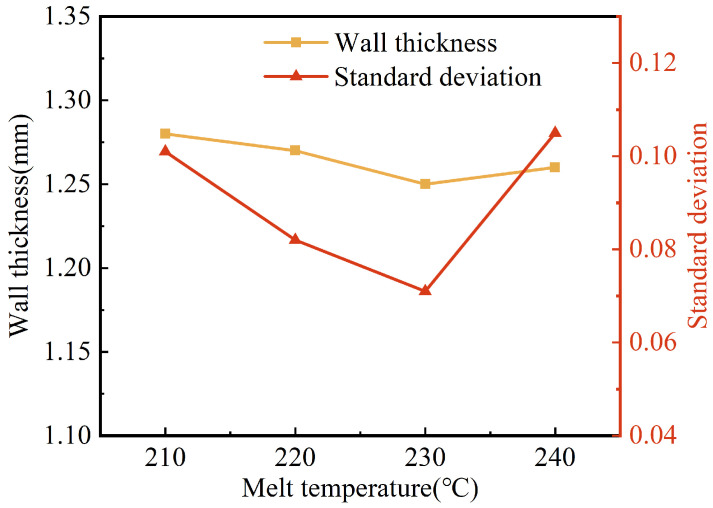
Effect of melt temperature on wall thickness and standard deviation of G-PAIM-O pipes.

**Figure 8 polymers-15-01985-f008:**
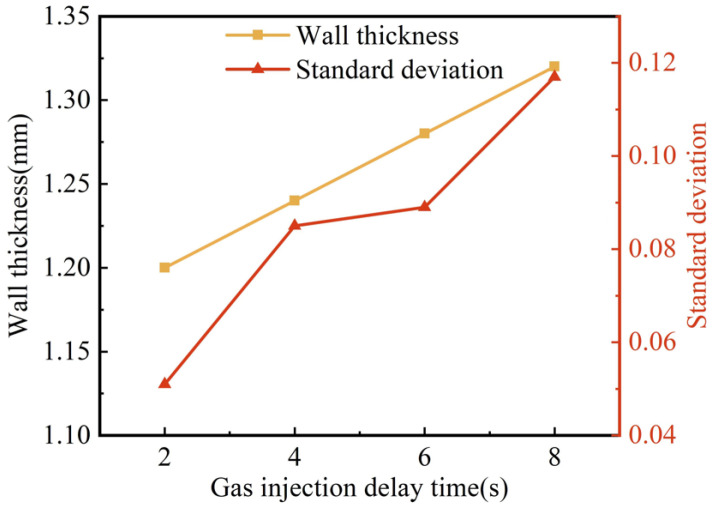
Effect of gas injection delay time on wall thickness and standard deviation of G-PAIM-O pipes.

**Figure 9 polymers-15-01985-f009:**
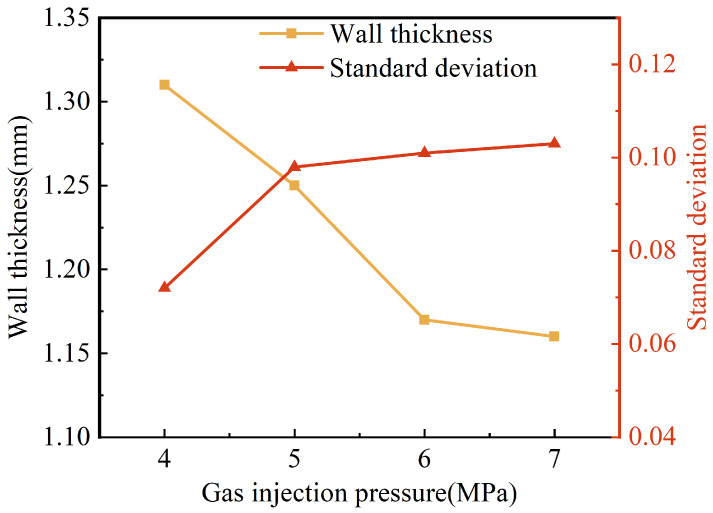
Effect of gas injection pressure on wall thickness and standard deviation of G-PAIM-O pipes.

**Figure 10 polymers-15-01985-f010:**
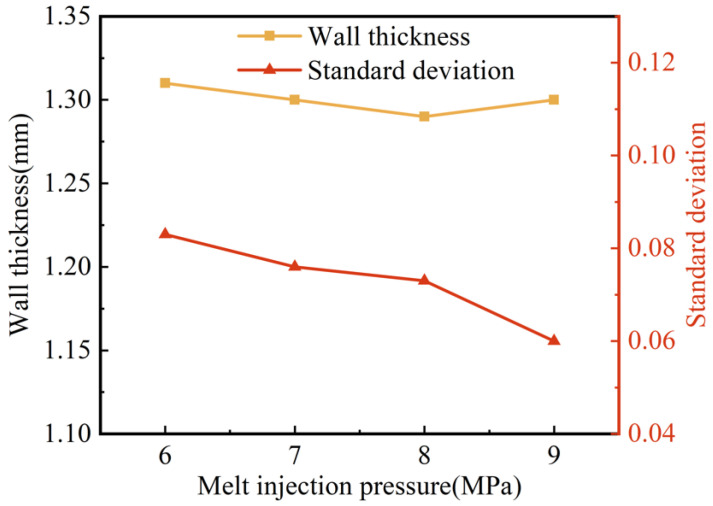
Effect of melt injection pressure on wall thickness and standard deviation of G-PAIM-O pipes.

**Figure 11 polymers-15-01985-f011:**
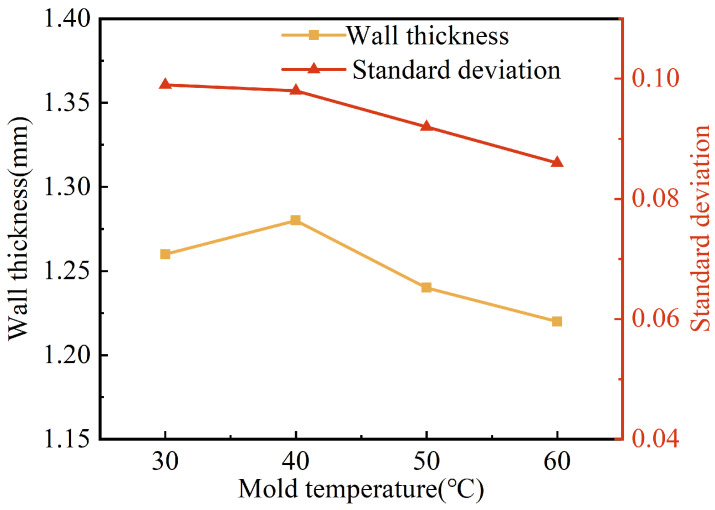
Effect of mold temperature on wall thickness and standard deviation of G-PAIM-O pipes.

**Table 1 polymers-15-01985-t001:** Processing parameters used in the experiments.

Processing Parameters	Levels			
	1	2	3	4
Melt temperature/°C	210	(220)	230	240
Gas injection delay time/s	2	(4)	6	8
Gas injection pressure/MPa	4	(5)	6	7
Melt injection pressure/MPa	6	(7)	8	9
Mold temperature/°C	30	(40)	50	60

## Data Availability

Not applicable.

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
