# Peer review of "Effects of Processing Method and Parameters on the Wall Thickness of Gas-Projectile-Assisted Injection Molding Pipes"

_polymers, 2023, doi:10.3390/polym15091985_

Round 1

Reviewer 1 Report (Previous Reviewer 1)

The Authors addressed all comments received previously. 

Author Response

Reply to reviewers’ comments for manuscript #polymers-2314439 entitled " Effects of Processing Method and Parameters on the Wall Thickness of Gas-Projectile-Assisted Injection Molding Pipes,"

We are deeply grateful for the reviewers’ comments and suggestions as well as the editorial suggestions. They helped to improve the quality of our paper and our current research. We have made the following changes to accommodate their comments and suggestions in the revision.

Reviewer 1:

The Authors addressed all comments received previously. 

Reply: We appreciate your positive comments and valuable suggestions to improve the quality of our manuscripts.

Reviewer 2 Report (Previous Reviewer 2)

The authors have addressed my comments. I find this fit for publication. 

Author Response

Reply to reviewers’ comments for manuscript #polymers-2314439 entitled " Effects of Processing Method and Parameters on the Wall Thickness of Gas-Projectile-Assisted Injection Molding Pipes,"

We are deeply grateful for the reviewers’ comments and suggestions as well as the editorial suggestions. They helped to improve the quality of our paper and our current research. We have made the following changes to accommodate their comments and suggestions in the revision.

Reviewer 2:

The authors have addressed my comments. I find this fit for publication.

Reply: We appreciate your positive comments and valuable suggestions to improve the quality of our manuscripts.

Reviewer 3 Report (New Reviewer)

The manuscript entitled "Effects of Processing Method and Parameters on the Wall Thickness of Gas-Projectile-Assisted Injection Molding Pipes" reports a study dealing with the formulation of PP pipes thorugh different injection molding processes and the evaluation of the process parameters on the wall thickness of the produced samples.

In my view, the manuscript could be considered for publication on Polymers after thet the following issues have been solved:

- the English language of the manuscript needs to be carefully revised in order to remove the typos present through the text;

- it is not clear why in the experimental part both straight and bend pipes were presented while the results of the study are reported only for the second ones;

- Figure 2 could be removed, as it does not report significant details;

- the Authors should specify how the levels of variation of the processing parameters reported in Table 1 were selected.

Author Response

Reply to reviewers’ comments for manuscript #polymers-2314439 entitled " Effects of Processing Method and Parameters on the Wall Thickness of Gas-Projectile-Assisted Injection Molding Pipes,"

We are deeply grateful for the reviewers’ comments and suggestions as well as the editorial suggestions. They helped to improve the quality of our paper and our current research. We have made the following changes to accommodate their comments and suggestions in the revision.

Reviewer 3:

The manuscript entitled "Effects of Processing Method and Parameters on the Wall Thickness of Gas-Projectile-Assisted Injection Molding Pipes" reports a study dealing with the formulation of PP pipes through different injection molding processes and the evaluation of the process parameters on the wall thickness of the produced samples.

In my view, the manuscript could be considered for publication on Polymers after that the following issues have been solved:

- the English language of the manuscript needs to be carefully revised in order to remove the typos present through the text;

Reply: We appreciate your positive comments and valuable suggestions to improve the quality of our manuscripts. In the revised manuscript, we have made detailed corrections to the English language of the manuscript, including typos, etc.

- it is not clear why in the experimental part both straight and bend pipes were presented while the results of the study are reported only for the second ones;

Reply: Thank you for your valuable comments. This is because bending pipes were used to investigate the effect of the process method on the wall thickness of the pipe, while straight pipes were used to investigate the effect of the processing parameters on the wall thickness of the Gas-Projectile-Assisted Injection molded pipe. It is introduced in the experimental scheme section of the revised manuscript. And the type of pipes used is presented in the Results and Discussion section.

- Figure 2 could be removed, as it does not report significant details;

Reply: Thank you for your valuable comments, it was our mistake. In the revised manuscript, we have removed Figure 2.

- the Authors should specify how the levels of variation of the processing parameters reported in Table 1 were selected.

Reply: Thank you again for your positive comments and valuable suggestions to help us improve the quality of the manuscript. The processing parameters and their levels in Table 1 were selected in conjunction with the recommendation of the material manufacturer as well as experimentally. Lines 129 to 130 in our revised manuscript describe how the process parameters and their levels were chosen.

Round 2

Reviewer 3 Report (New Reviewer)

I recommend the publication of the paper as it stands

This manuscript is a resubmission of an earlier submission. The following is a list of the peer review reports and author responses from that submission.

Round 1

Reviewer 1 Report

The main concentration of this paper is on Gas-Projectile-Assisted Injection Molding and Gas-Assisted Injection Molding. The comparison between these two methods has been conducted in terms of wall thickness of the pipes and its uniformity. Different process parameters have been assigned to determine the effects of individual parameters on wall thickness of the selected product. The research shows that the effect of f gas injection delay time and gas injection pressure on the wall thickness of G-PAIM-O pipes were significant.

To improve the quality of the paper, please apply the following comments:

1.       In line 9, you mentioned the short shot method. More explanation is needed to add to the paper about this method as short shot is not a method but also an external defect in injection molding.

2.       In line 121: The experimental method (the single-factor experimental method) is used to run the trials based on the selected parameters on Table 1. More details are needed to explain how you collect the data set or how you run each trial. Did you run the experiments using Taguchi method orthogonal array? If yes, then you need to include the details that which quality characteristics had been applied.

Author Response

Reply to reviewers’ comments for manuscript #polymers-2162936 entitled " Effects of Processing Method and Parameters on the Wall Thickness of Gas-Projectile-Assisted Injection Molding Pipes,"

We are deeply grateful for the reviewers’ comments and suggestions as well as the editorial suggestions. They helped to improve the quality of our paper and our current research. We have made the following changes to accommodate their comments and suggestions in the revision.

Reviewer 1:

Comments and Suggestions for Authors:

The main concentration of this paper is on Gas-Projectile-Assisted Injection Molding and Gas-Assisted Injection Molding. The comparison between these two methods has been conducted in terms of wall thickness of the pipes and its uniformity. Different process parameters have been assigned to determine the effects of individual parameters on wall thickness of the selected product. The research shows that the effect of f gas injection delay time and gas injection pressure on the wall thickness of G-PAIM-O pipes were significant.

To improve the quality of the paper, please apply the following comments:

  1. In line 9, you mentioned the short shot method. More explanation is needed to add to the paper about this method as short shot is not a method but also an external defect in injection molding.

Reply: Thank you for your valuable comments. In conventional injection molding process, short shot is a defect that the cavity is not filled and the part is incomplete. While in fluid-assisted injection molding process, short shot is one of its variants that the cavity is not filled completely before the fluid injection. More detail explanation of the Short-shot method of Gas-Projectile Assisted Injection Molding process was added in the revised manuscript.

  1. In line 121: The experimental method (the single-factor experimental method) is used to run the trials based on the selected parameters on Table 1. More details are needed to explain how you collect the data set or how you run each trial. Did you run the experiments using Taguchi method orthogonal array? If yes, then you need to include the details that which quality characteristics had been applied.

Reply: Thank you for your valuable comments. The available processing window was determined via combining the recommendation of material manufacturer and experiments. Then the processing parameters were set. While Taguchi method was not used in this research, it will be adopted in the experiments design to optimize the process in our future work.

Reviewer 2 Report

The authors have systematically investigated the effects of process methods, including Short-shot Gas-Projectile Assisted Injection Molding (G-PAIM-S), Overflow Gas-Projectile Assisted Injection Molding (G-PAIM-O), Short-shot Gas-Assisted Injection Molding (GAIM-S) and Overflow Gas-Assisted Injection Molding (GAIM-O), on the wall thickness and wall thickness uniformity of pipes. Additionally, they used the single-factor method to investigate the influence of processing parameters on the 80 wall thickness and wall thickness uniformity of G-PAIM-O pipes.

The conception, execution and discussion of this work is very good and the experiments are conducted systematically with appropriate discussion. This work is fit for publishing pending these minor changes:

1) Figures 7 to 11: I am not sure why the authors are plotting error bars on a separate y axis ? Can they not put error bars on the existing curves ?

2) The conclusion sections has both these statements: 

"The influence of the gas injection pressure and gas injection delay time on the wall  thickness of G-PAIM-O pipes were significant, while those of the melt temperature, melt  injection pressure and mold temperature on it were minimal. The longer the gas injection  delay time and the lower the gas injection pressure, the thicker the wall thickness is. 

The wall thickness uniformity of G-PAIM-O pipes was good and less affected by the processing parameters."

These 2 statements contradict each other. Please revise. 

Author Response

Reply to reviewers’ comments for manuscript #polymers-2162936 entitled " Effects of Processing Method and Parameters on the Wall Thickness of Gas-Projectile-Assisted Injection Molding Pipes,"

We are deeply grateful for the reviewers’ comments and suggestions as well as the editorial suggestions. They helped to improve the quality of our paper and our current research. We have made the following changes to accommodate their comments and suggestions in the revision.

Comments and Suggestions for Authors:

The authors have systematically investigated the effects of process methods, including Short-shot Gas-Projectile Assisted Injection Molding (G-PAIM-S), Overflow Gas-Projectile Assisted Injection Molding (G-PAIM-O), Short-shot Gas-Assisted Injection Molding (GAIM-S) and Overflow Gas-Assisted Injection Molding (GAIM-O), on the wall thickness and wall thickness uniformity of pipes. Additionally, they used the single-factor method to investigate the influence of processing parameters on the 80 wall thickness and wall thickness uniformity of G-PAIM-O pipes.

The conception, execution and discussion of this work is very good and the experiments are conducted systematically with appropriate discussion. This work is fit for publishing pending these minor changes:

1) Figures 7 to 11: I am not sure why the authors are plotting error bars on a separate y axis ? Can they not put error bars on the existing curves ?

Reply: Thank you for your valuable comments. In this paper, the standard deviation of wall thicknesses was calculated to indicate the wall thickness uniformity. In order to show the influence of process method and processing parameters on the wall thickness uniformity more visually, the standard deviation were plotted on a separate y axis.

2) The conclusion sections has both these statements:

"The influence of the gas injection pressure and gas injection delay time on the wall thickness of G-PAIM-O pipes were significant, while those of the melt temperature, melt injection pressure and mold temperature on it were minimal. The longer the gas injection delay time and the lower the gas injection pressure, the thicker the wall thickness is.

The wall thickness uniformity of G-PAIM-O pipes was good and less affected by the processing parameters."

These 2 statements contradict each other. Please revise.

Reply: Thank you again for your valuable comments. The latter statement means among the four variants of Gas-Assisted Injection Molding process, the wall thickness uniformity of G-PAIM-O pipes was good and less affected by the processing parameters. While in the G-PAIM-O process, the gas injection pressure and gas injection delay time are the main influence factors of the wall thickness, as the former statement described. In order to eliminate the possible misunderstandings, a minor revision has made in the manuscript.

Reviewer 3 Report

The paper is devoted to the important problem.

However, I have one serious problem with this manuscript.

There is no physical model explaining the process of gas-projectile-assisted injection, reported in the manuscript. This fact diminishes the scientific soundness of the paper essentially. The authors should supply such a model in the revised version of the manuscript.

Author Response

Reply to reviewers’ comments for manuscript #polymers-2162936 entitled " Effects of Processing Method and Parameters on the Wall Thickness of Gas-Projectile-Assisted Injection Molding Pipes,"

We are deeply grateful for the reviewers’ comments and suggestions as well as the editorial suggestions. They helped to improve the quality of our paper and our current research. We have made the following changes to accommodate their comments and suggestions in the revision.

Comments and Suggestions for Authors:

The paper is devoted to the important problem.

However, I have one serious problem with this manuscript.

There is no physical model explaining the process of gas-projectile-assisted injection, reported in the manuscript. This fact diminishes the scientific soundness of the paper essentially. The authors should supply such a model in the revised version of the manuscript

Reply: Thank you for your valuable comments. Figure 1 in the manuscript can be used to explain the process of Gas-Projectile Assisted Injection Molding detailly if the fluid in it is nitrogen. Its overflow process is described from line 50 to line 60. Photos of the projectile used in the experiments and longitudinal profile of samples molded by different processes are added in Figure. 2b and Figure. 5, respectively. They provide a better understanding of the Gas-Projectile Assisted Injection Molding process.

Round 2

Reviewer 3 Report

The authors completely ignored the remarks suggested by the reviewer. The paper contains no physical model supporting the conclusions made by the authors. The paper should be rejected.

Author Response

Reply to reviewers’ comments for manuscript #polymers-2162936 entitled " Effects of Processing Method and Parameters on the Wall Thickness of Gas-Projectile-Assisted Injection Molding Pipes,"

We are deeply grateful for the reviewers’ comments and suggestions as well as the editorial suggestions. They helped to improve the quality of our paper and our current research. We have made the following changes to accommodate their comments and suggestions in the revision.

Reviewer 3:

Comments and Suggestions for Authors

The authors completely ignored the remarks suggested by the reviewer. The paper contains no physical model supporting the conclusions made by the authors. The paper should be rejected.

Reply: We appreciate your valuable comments. We are sorry that we did not understand your point during the last round of changes. We have not overlooked the issues you raised. We kindly ask you to give us another chance, and now we have done our best to make the changes. We have provided a complete physical model of the Gas-Projectile Assisted Injection Molding process as shown in Figure 2. Combined with Figure 1 and Figure 3, it can help us understand the molding process of Gas-Projectile Assisted Injection Molding process better. We appreciate for Reviewers' warm work earnestly and hope that the correction will meet with approval. Finally, we sincerely apologize again. Thank you for your help. We wish you good luck in your work and happy life.

Round 3

Reviewer 3 Report

The paper was not improved and should be rejected.